# Signs Indicative of Central Sensitization Are Present but Not Associated with the Central Sensitization Inventory in Patients with Focal Nerve Injury

**DOI:** 10.3390/jcm11041075

**Published:** 2022-02-18

**Authors:** Luis Matesanz-García, Ferran Cuenca-Martínez, Ana Isabel Simón, David Cecilia, Carlos Goicoechea-García, Josué Fernández-Carnero, Annina B. Schmid

**Affiliations:** 1Escuela Internacional de Doctorado, Department of Physical Therapy, Occupational Therapy, Rehabilitation and Physical Medicine, Rey Juan Carlos University, 28922 Alcorcón, Spain; luis.matesanzgarcia@gmail.com; 2Department of Physiotherap, Centro Superior de Estudios Universitarios La Salle, Universidad Autónoma de Madrid, 28023 Madrid, Spain; 3Exercise Intervention for Health Research Group (EXINH-RG), Department of Physiotherapy, University of Valencia, 46010 Valencia, Spain; fecuen2@gmail.com; 4Unit of Elbow-Hand, Service de Traumatología, Hospital Severo Ochoa, 28911 Leganés, Spain; ai.simoncarrascal@gmail.com; 5Unit of Elbow-Hand, Service de Traumatología, Hospital 12 de Octubre, 28048 Madrid, Spain; dacecilia@hotmail.com; 6Complutense University of Madrid, 28040 Madrid, Spain; 7Department of Surgery, Hospital Vithas La Milagrosa, 28010 Madrid, Spain; 8Department Basic Health Sciences, Rey Juan Carlos University, 28922 Alcorcón, Spain; carlos.goicoechea@urjc.es; 9Grupo Multidisciplinar de Investigación y Tratamiento del Dolor, Grupo de Excelencia Investigadora URJC-Banco de Santander, 28922 Madrid, Spain; 10Department of Physical Therapy, Occupational Therapy, Rehabilitation and Physical Medicine, Rey Juan Carlos University, 28922 Alcorcón, Spain; 11Nuffield Department of Clinical Neurosciences, University of Oxford, Oxford OX3 9DU, UK

**Keywords:** entrapment neuropathy, conditioned pain modulation, temporal summation, pain measurement, carpal tunnel syndrome, pressure pain threshold, central sensitization, central sensitization inventory

## Abstract

Objective: Carpal tunnel syndrome (CTS) is the most common focal nerve injury. People with CTS may show alterations in central processing of nociceptive information. It remains unclear whether the central sensitization inventory (CSI) is capable of detecting such altered central pain processing. Methods: Thirty healthy volunteers were matched with 30 people with unilateral CTS from the orthopaedic waitlist. Changes to central pain processing were established through psychophysical sensory testing (bilateral pressure pain thresholds (PPT), conditioned pain modulation, temporal summation) and pain distribution on body charts. Patients also completed pain severity and function questionnaires, psychological questionnaires and the CSI. Results: Compared to healthy volunteers, patients with CTS have lower PPTs over the carpal tunnel bilaterally (t = −4.06, *p* < 0.0001 ipsilateral and t = −4.58, *p* < 0.0001 contralateral) and reduced conditioned pain modulation efficacy (t = −7.31, *p* <0.0001) but no differences in temporal summation (t = 0.52, *p* = 0.60). The CSI was not associated with psychophysical measures or pain distributions indicative of altered central pain processing. However, there was a correlation of the CSI with the Beck Depression Inventory (r = 0.426; *p* = 0.019). Conclusion: Patients with CTS show signs of altered central pain mechanisms. The CSI seems unsuitable to detect changes in central pain processing but is rather associated with psychological factors in people with focal nerve injuries.

## 1. Introduction

Carpal tunnel syndrome (CTS) is the most common focal nerve injury [1,2]. It is defined as a compression of the median nerve as it passes through the carpal tunnel in the wrist. Classically, CTS symptoms manifest themselves in the median nerve area, although extramedian or proximal spread of symptoms is frequently reported [3]. This spread of symptoms has been attributed to changes in the central nervous system such as central sensitization [4].

Central sensitization is a neurophysiological mechanism that cannot be directly determined in humans. However, in addition to the spread of symptoms, psychophysical sensory testing can be used to infer the contribution of central pain mechanisms to patients’ presentations. For instance, local and remote mechanical hyperalgesia such as measured with pain thresholds has been associated with central sensitization [5,6]. Additionally, temporal summation is related to activity-dependent plasticity within the central nervous system [7,8]. Several studies have examined the presence of such local and widespread hyperalgesia in patients with CTS with conflicting results [9,10].

In addition to increased facilitation, a disruption of inhibitory mechanisms is another central mechanism that can lead to hyperexcitability. Conditioned pain modulation (CPM) is a psychophysical measure that examines the efficacy of endogenous inhibitory systems [11]. CPM evaluates whether a painful test stimulus can be modulated by a noxious conditioning stimulus applied at a remote part of the body. About 70% of patients with chronic pain show signs of reduced CPM efficacy [12]. To date, only one publication has examined possible alterations of the descending inhibitory system by means of CPM in patients with CTS and found reduced efficacy [13]. 

Whereas these psychophysical sensory tests can provide information about the potential involvement of central pain mechanisms, they are time consuming and involve costly equipment. Thus, self-completed questionnaires have been developed to identify the presence of “central sensitization”. For instance, the central sensitization inventory (CSI) has been suggested to identify patients with “central sensitivity syndrome” such as fibromyalgia, chronic fatigue syndrome, irritable bowel or temporomandibular joint disorders [14]. In addition, the CSI is associated with outcomes after spinal surgery [15]. However, recent studies question its construct validity. The CSI was originally validated by demonstrating increasing CSI scores in conditions thought to represent increasing degrees of central sensitization (e.g., healthy controls, regional chronic low back pain, chronic widespread pain, fibromyalgia) [14]. Similarly, the cutoff to identify “central sensitivity syndrome” was determined by receiver operating curve analyses, best distinguishing patients with diagnoses that are thought to be characterised by central mechanisms (e.g., fibromyalgia) from healthy controls [16]. Arguably, a more compelling way of evaluating the construct validity of a tool that is meant to identify central sensitisation is to examine its associations with psychophysical testing. However, recent studies in patients with temporomandibular disorders, shoulder pain, chronic whiplash and chronic spinal pain found no relationship between the CSI and psychophysical tests indicating the presence of central pain mechanisms [17,18,19,20]. In contrast, other studies have identified a weak correlation of the CSI with mechanical hyperalgesia and CPM in patients with knee osteoarthritis [21], and CSI scores seem higher in patients with musculoskeletal pain and more impaired CPM [22]. Of note, evidence is growing that the CSI is more strongly associated with psychological measures rather than psychophysical measures indicating central pain mechanisms [18,19,20,21,23]. To date, construct validity of the CSI has only been evaluated in populations with musculoskeletal pain. It remains unclear how the CSI performs in patients with peripheral nerve injuries. 

To improve our knowledge of alterations in pain processing in focal nerve injuries and the construct validity of the CSI, this study has the following objectives: (1) Identify alterations in central pain mechanisms in patients with CTS using psychophysical sensory testing and pain mapping. (2) Investigate whether the CSI is associated with psychophysical parameters and pain distributions indicative of central pain mechanisms. (3) Investigate whether the CSI is associated with psychological parameters.

## 2. Materials and Methods

### 2.1. Participants

Thirty patients with unilateral CTS were recruited from Hand and Elbow surgery units from 12 Octubre Hospital and Severo Ochoa Hospital, both located in Madrid. All patients were on the orthopaedic surgery waitlist, with at least one year of persistency of symptoms, had positive Tinel’s and Phalen’s sign and had electrodiagnostic confirmation of moderate to severe CTS on the affected side according to the American Association of Neuromuscular and Electrodiagnostic Medicine [24]. Patients were excluded if the electrodiagnostic testing identified sensory and/or motor deficits of the radial and/or ulnar nerve, if any indication for nerve root involvement was present (e.g., needle EMG) or if patients reported previous hand surgeries, previous steroid infiltrations, wrist fractures, diagnoses related to the cervical spine and upper limb (e.g., cervical radiculopathies, shoulder injuries), or other musculoskeletal comorbidities (e.g., rheumatoid arthritis and fibromyalgia). Women who were pregnant were excluded from the study. 

The patients were matched for age and sex with healthy controls (HC, n = 30). Those were recruited through advertisements around the hospitals and university and through relatives of participating patients. All participants gave informed written consent prior to participating, and the study received ethical clearance from the two committees of the participating hospitals CPMP/ICH/135/95 (Severo Ochoa Hospital, December 2017) and 20/092 (12 Octubre Hospital, March 2020).

### 2.2. Symptom Characteristics and Functional Deficits

The Boston carpal tunnel questionnaire was used to assess symptom severity and functional deficits. The Boston carpal tunnel questionnaire consists of a symptom and a function subscale [25]. This questionnaire has been validated in Spanish, with good levels of internal consistency and reproducibility [26]. The current hand pain intensity was recorded using a visual analogue scale (VAS), with 0 being no pain and 100 being the worst pain imaginable. 

The presence of neuropathic pain was assessed with the Spanish version of Douleur Neuropathique 4 (DN4). This version has shown good internal consistency [27]. The questionnaire consists of an initial part with questions that evaluate a series of neuropathic symptom descriptors (burning and cold-like pain, electric shock, tingling, pins and needles, numbness, itching) followed by a short sensory clinical examination (hypoesthesia to touch, hypoesthesia to pin prick and brush allodynia). A DN4 score of ≥4 was interpreted as neuropathic pain [28] and a score < 4 was interpreted as nociceptive pain. 

The central sensitization inventory (CSI) is a tool originally designed to identify patients with “central sensitivity syndrome” [14]. It includes a wide range of 25 questions covering pain and stiffness, daily function, psychological factors (e.g., anxiety, depression), fatigue and memory. The Spanish translation shows good reliability and internal consistency [29]. The patient scores each question from 0 to 4: never, rarely, sometimes, continuously and always. The total score ranges from 0 to 100 with values over 40 thought to indicate the presence of “central sensitivity syndrome” [14]. 

### 2.3. Signs Indicative of Altered Central Pain Processing

#### 2.3.1. Pressure Pain Threshold (PPT) 

PPT is defined as the minimum amount of pressure needed to elicit pain. Measurements of PPT were made using a digital algometer (Model FDX 10^®^, Wagner Instruments, Greenwich, CT, USA). This instrument measures the pressure in kg/cm^2^. The measurements were made bilaterally (affected and unaffected side) over the carpal tunnel (Appendix A). The average of three measurements was recorded, with an interval of 30 s between each measurement to avoid a temporal summation effect. PPT has shown good reliability and internal consistency [30]. 

#### 2.3.2. CPM

For the evaluation of CPM efficiency, an average of three PPT measures was used as a test stimulus over the base of the dorsal side of the distal phalanx of the thumb of the affected side (Appendix A). The conditioning stimulus involved ischemic pain using a sphygmomanometer applied on the unaffected arm with a pressure of 200 mmHg until the subjects reported pain intensity between 5–7/10 on a numerical pain rating scale. While the sphygmomanometer was still inflated, the PPT measurements were repeated on the dorsal side of the distal phalanx of the thumb on the affected side. This protocol has been shown to be adequate to assess the endogenous inhibitory system in patients with knee osteoarthritis [31]. CPM efficacy was calculated by deducting the PPT after applying the conditioning stimulus from the PPT obtained before the conditioning stimulus. Positive values indicate effective pain modulation [32]. 

#### 2.3.3. Temporal Summation

The measurement of temporal summation was performed using a Model FDX 10^®^ digital algometer, Wagner Instruments, Greenwich, CT, USA applied to the intensity of the PPT at the midpoint between the nail and the interphalangeal joint on the dorsal side of the distal phalanx of the first finger of the affected side (Appendix A). Numerical pain ratings from 0–10 were obtained for a single stimulus followed by a rating after 10 stimuli with a repetition rate of 1 Hz. For the isolated stimuli, the patients were asked to indicate the onset of pain and rate it from 0–10. The repetitive stimuli were performed in an area around the same point of the finger with the same pressure that induced the first onset of pain during the isolated stimulus. The average pain intensity after 10 repetitions was recorded. The temporal summation ratio was calculated by dividing the average pain produced by the train of stimuli by the pain produced by the single stimulus. A similar method has been used and validated previously [33]. 

#### 2.3.4. Symptom Spread

Patients marked the localization of symptoms on a hand and body diagram [34]. Results were dichotomized into median and extramedian distribution. 

### 2.4. Emotional Wellbeing

To evaluate emotional wellbeing, patients completed the Beck Depression Inventory (BECK) and the State-Trait Anxiety Inventory (STAI). The BECK consists of 21 elements related to depressive symptoms (e.g., hopelessness and irritability), specific thoughts (e.g., guilt or feelings of being punished) and physical symptoms [35,36]. STAI has demonstrated acceptable psychometric properties in its Spanish version [37].

To assess pain-related fear of movement, the validated Tampa kinesiophobia scale (TSK) was used. Each item is rated on a four-point Likert scale ranging from “strongly agree” to “strongly disagree” with a cutoff of 29 points. This questionnaire has shown a good consistency [38,39].

### 2.5. Statistical Analyses

The sample size was estimated using the program G*Power 3.1.7 (G*power from University of Dusseldorf, Germany) [40]. The sample size calculation was powered to detect between-group differences in PPT measures. Using previously published data measured over the carpal tunnel area in healthy controls and patients with CTS [41], n = 30 participants are required in each group to detect an effect size of 0.74 with 80% statistical power (alpha = 0.05, independent *t*-test). This sample size is sufficient to detect large effects in correlation analyses (rho = 0.44, power 80%, alpha 0.05).

We performed the data analysis using the Statistics Package for Social Science (SPSS 20.00, IBM Inc., Armonk, NY, USA). We checked data normality by visual inspection of histograms and the Kolmogorov–Smirnov test. Participants’ sociodemographic and clinical characteristics were summarized using descriptive statistics and summary tables.

To determine the presence of signs indicative of altered central pain processing, we employed independent Student’s *t*-tests to identify differences between healthy and patient groups for psychophysical variables. The frequency of extraterritorial spread of symptoms (median/extramedian) was reported.

To identify associations between CSI and psychophysical signs indicative of altered pain processing, we performed Pearson’s correlation statistics in patient data only. Coefficients of 0.5 or above were interpreted as a strong correlation, 0.3 moderate and 0.1 small correlation. We corrected for false discovery rate using the Benjamini–Hochberg correction (FDR = 25%). We also grouped patients with CTS into those with CSI ≥ 40 and <40 and explored differences in psychophysical tests and symptom spread with independent *t*-tests and Chi squared or Fisher’s exact test statistics as appropriate.

To explore the relationship between CSI and emotional wellbeing (BECK, TSK, STAI) in patient data, we calculated Pearson correlations coefficients and used Benjamini–Hochberg correction to correct for a false discovery rate. Unadjusted *p*-values are reported for ease of interpretation.

## 3. Results

Thirty participants were healthy controls (8 men and 22 women with a mean age of 46.23 ± 1.36 years), and 30 patients diagnosed with CTS (8 men and 22 women with a mean age of 48.67 ± 1.19 years, Table 1). According to the Boston questionnaire, patients had on average mild to moderate symptoms and moderate to severe function deficits. The mean pain intensity was 4.2/10 (SD 2.7). Using the DN4 questionnaire, the most common pain descriptor was tingling (100%), followed by numbness (96.6%) and electric shocks (90%). In contrast, hypoesthesia to touch and pinprick was present only in 50% and 60% of patients, respectively. Twenty-eight patients (93.3%) were classified as having neuropathic pain according to the DN4.

### 3.1. Patients with CTS Have Signs Indicative of Altered Central Pain Processing

There were statistically significant differences between patients with CTS and healthy participants in the psychophysical variables related to central pain processing. PPTs were reduced in patients with CTS, indicative of mechanical hyperalgesia compared to healthy controls both on the ipsilateral (t = −4.06; *p* < 0.0001) and contralateral side (t = −4.58; *p* < 0.0001).

Similarly, CPM efficiency was reduced in patients with CTS compared to healthy controls (t = −7.31; *p* < 0.01). No differences were found for temporal summation (t = 0.52, *p* = 0.60). Data are shown in Table 2.

### 3.2. Association between Central Sensitization Inventory and Signs of Altered Central Pain Processing

The mean CSI in patients with CTS was 32.4 (SD 11.8). Eight patients (26.67%) had a score ≥ 40.

The CSI did not correlate with any psychophysical signs of altered pain processing (Table 3). Similarly, there were no differences in psychophysical signs of altered pain processing if patients were grouped according to the CSI cutoff of ≥40 (*p* > 0.600).

Extramedian distribution of symptoms was reported by 25 (83%) of patients with the remaining patients reporting a median distribution. No difference was identified for the proportion of patients with median/extramedian spread of symptoms according to the CSI cutoff (Table 4).

### 3.3. Association between Central Sensitization Inventory and Emotional Wellbeing

The mean BECK, STAI and TSK scores in patients with CTS were 7.87 (SD 4.91), 24.30 (SD 5.05), and 25.93 (SD 7.62), respectively (Appendix A). The CSI did not correlate with the level of anxiety according to STAI (r = 0.026; *p* = 0.893) and kinesiophobia according to TSK (r = 0.109; *p* = 0.566). There was, however, a moderate correlation between CSI and depression according to BECK (r = 0.426; *p* = 0.019, Table 5), which survived the Benjamini–Hochberg correction.

## 4. Discussion

Our cohort of patients with CTS has clear indications for the presence of central pain mechanisms as apparent by local and widespread mechanical hyperalgesia and impaired CPM compared to healthy volunteers. No changes were apparent for temporal summation. Of note, there was no association of the CSI with psychophysical measures or symptom spread indicative of central pain mechanisms. There was, however, a moderate correlation between the CSI and depression scores, suggesting that the CSI may be more closely related to psychological parameters than psychophysical measures indicative of central pain mechanisms in patients with focal nerve injury.

Our cohort of patients with CTS had clear indication of a presence of central pain mechanisms, although there was heterogeneity among patients. We identified mechanical hyperalgesia both locally as well as remotely, extraterritorial spread of symptoms and lower efficacy of CPM. Extraterritorial spread of symptoms in patients with CTS is consistently reported in the literature [4,42,43] and has been associated with the presence of central mechanisms. Mechanical hyperalgesia is also commonly interpreted as a sign of central sensitization [44,45]. Local (and remote) mechanical hyperalgesia has previously been reported in focal peripheral neuropathies including CTS [9,43,46]. However other patient cohorts could not confirm this at group level [47,48]. This discrepancy may be attributed to different recruitment pathways as well as different sites of PPT measurements (e.g., palmar aspect of index finger, carpal tunnel). Importantly, the large variation in mechanical hyperalgesia within patients with CTS suggests differing extents of central contributions in individual patients.

Intriguingly, this is the second study demonstrating impaired CPM efficacy in patients with CTS (see also Soon et al., 2017) [13]. On the other hand, temporal summation, which is related to activity-dependent plasticity within the central nervous system [7,8] remained unaltered in patients with CTS. Whereas we assessed temporal summation with PPTs as used in other cohorts [33], more established protocols using pinprick stimulators also did not find group differences between patients with CTS and healthy participants [47]. In line with our results, temporal summation is often found to be comparable at group level in other peripheral neuropathies including systemic polyneuropathies [49,50] and other focal nerve injuries [46,51]. Again though, there is variation within patients, suggesting that some patients have elevated temporal summation, which may be washed out in group comparisons.

Of note, we did not identify an association between the CSI and psychophysical measures and symptom location indicative of altered central pain processing. The CSI was originally developed as a tool to identify central sensitization characteristics [52]. It was developed in patients with fibromyalgia, chronic widespread pain and chronic low back pain, who presumably have stronger clinical phenotypes than the here studied patients with entrapment neuropathies. Some studies in musculoskeletal pain report a correlation between the CSI and the spread of pain [21,53]. However, similar to our findings in patients with peripheral nerve injury, other studies do not find a correlation of the CSI with symptom spread in people with shoulder pain [20] and whiplash injury [18], questioning its construct validity.

A recent systematic review reports a high construct validity of the CSI [54]. However, the included studies compared the CSI to other questionnaires related to pain severity, general health, emotional wellbeing or sleep. There may be a reciprocal relationship of these measures with central pain mechanisms. However, these constructs are not measures of central sensitisation, which the CSI is meant to evaluate. Surprisingly, not even the original development of the CSI involved psychophysical measures of central pain mechanisms, which are considered to be best practice when assessing the manifestation of central sensitisation in humans [55]. Recent studies have compared the CSI with psychophysical tests indicative of central pain mechanisms. Of note, most studies find no [17,18,19] or only a weak correlation [21,53] between the CSI and psychophysical tests in patients with musculoskeletal pain. This, together with our findings of no association between the CSI and psychophysical tests in patients with focal nerve injury, further questions the validity of the CSI in detecting human correlates of central sensitization.

Intriguingly though, the CSI was associated with depressive symptoms determined on the BECK in our cohort. Such an association of CSI with psychological wellbeing has been consistently reported in the literature [17,18,19,20,21,23]. This may not be surprising as several questions of the CSI explore psychological constructs such as anxiety, feeling sad or depressed. Whereas, indeed, a decrease in emotional wellbeing is frequently associated with chronic pain including neuropathic pain [46,49,50,56,57,58], care has to be taken to not confuse changes in emotional wellbeing with the presence of central sensitisation [59]. Unfortunately, these two distinct principles are often equated in the clinical literature. We should note though that whereas psychological parameters were more pronounced in patients with CTS compared to healthy volunteers in our study, average scores were not considered clinically relevant. These findings are in line with previous reports in patients with CTS [42].

### 4.1. Limitations

Some limitations have to be taken into account. The sample size was calculated to detect differences in central pain processing between healthy people and patients with CTS. Whereas it was adequately powered to detect large effects on correlations between the CSI and psychophysical testing, small or moderate correlations would have been missed. Inspection of the data, however, clearly demonstrated the absence of trends, and even if larger samples may have detected significant correlations, these would likely have been weak.

We recruited patients from surgery waiting lists, which is likely to include more severe profiles. However, symptom and function severity in our study was comparable with previous CTS cohorts from primary care [60] and secondary care [42]. The examiner who performed psychophysical testing could not be blinded to group allocation (CTS vs. healthy). To minimise bias, the examiner was not aware of the outcome of the CSI and other questionnaires until after psychophysical testing was performed. As per routine practice in participating hospitals, the electrodiagnostic test was only performed on the affected side. Subclinical cases of CTS on the contralateral side may therefore have been missed [61].

### 4.2. Clinical Implications

Our study confirms the presence of central pain mechanisms in patients with focal nerve injury. This is of clinical relevance as their presence may be associated with poorer prognosis in some musculoskeletal conditions [62]. It has also been suggested that the identification of central pain mechanisms may help personalise management strategies [63], an area of active research. For instance, duloxetine may be particularly effective in patients with peripheral diabetic neuropathy who have altered CPM efficacy [64]. Similarly, CPM efficacy may predict the analgesic effect of non-steroidal antirheumatic inflammatory drugs plus acetaminophen in patients with knee osteoarthritis [65], and temporal summation seems to predict pain relief from ketamine in patients with neuropathic pain [66]. Future studies will have to examine whether the identification of central pain mechanisms may be important not only for pharmacological management but also beyond (e.g., physiotherapy).

Most studies of personalised management according to central pain mechanisms use time-consuming psychophysical testing. A low-cost self-reported questionnaire that identifies central sensitisation as measured by psychophysical tools would be ideal. Unfortunately, our study adds to the increasing body of evidence that questions the usefulness of the CSI in identifying central sensitisation according to psychophysical measures. Rather, our data, together with other studies, consistently suggest that the CSI better reflects emotional wellbeing. It is crucial that the distinct concept of emotional wellbeing is not conflated with the neurophysiological concept of central sensitisation in clinical practice. Nevertheless, even though the CSI may not be detecting “central sensitisation” in a strict sense, it may still be of value clinically. For instance, CSI scores seem to be associated with prognostic outcome in certain musculoskeletal conditions [67,68], and this could be further explored in focal nerve injuries. 

## 5. Conclusions

Our results suggest that patients with CTS have changes indicative of altered central pain processing. The CSI does not seem to be associated with psychophysical measures of central sensitization. Rather, the CSI correlates with emotional wellbeing, in particular, depression scores. These data question the construct validity of the CSI in detecting central sensitisation in patients with focal peripheral nerve injury.

## Figures and Tables

**Table 1 jcm-11-01075-t001:** Participant characteristics.

	Healthy (n = 30)	CTS (n = 30)
Female, n (%)	22 (77.3)	22 (77.3)
Age (Years)	46.2 ± 1.36	48.7 ± 1.2
Boston		
Severity		2.6 ± 0.11
Functional deficits		3.5 ± 0.11
Visual Analogue Scale		4.2 (2.7)
DN4 total score		5.9 (1.6)
Burning, n (%)		11 (36.6)
Painful cold, n (%)		9 (30.0)
Electric shocks, n (%)		27 (90.0)
Tingling, n (%)		30 (100.0)
Pins and needles, n (%)		22 (73.3)
Numbness, n (%)		29 (96.6)
Itching, n (%)		9 (30.0)
Hypoesthesia to touch, n (%)		15 (50.0)
Hypoesthesia to pinprick, n (%)		18 (60.0)
DN4 neuropathic, n (%)		28 (93.3)

Data are shown as mean and standard deviation or n (%).

**Table 2 jcm-11-01075-t002:** Variables indicative of changes in central pain processing.

	Healthy (n = 30)	CTS (n = 30)	*p*-Value
PPT affected side (Kg/cm^2^)	5.9 (2.0)	3.4 (1.7)	*p* < 0.0001
PPT contralateral side (Kg/cm^2^)	5.9 (2.0)	3.8 (2.0)	*p* < 0.0001
CPM	2.1 (2.0)	0.1 (0.9)	*p* < 0.0001
Temporal summation ratio	1.5 (0.9)	1.6 (0.9)	*p* = 0.60

Data are shown as mean (standard deviation); PPT: pressure pain threshold; CPM: conditioned pain modulation; *p*-values reflect Student’s *t*-tests.

**Table 3 jcm-11-01075-t003:** Correlations between CSI and signs of altered central pain processing in patients with CTS.

	Pearson Correlation Coefficient	Unadj *p*-Value
CSI vs. PPT affected side	0.023	0.903
CSI vs. PPT unaffected side	−0.042	0.828
CSI vs. CPM	0.276	0.140
CSI vs. temporal summation	0.069	0.719

CSI: central sensitization inventory; PPT: pain pressure threshold; CPM: conditioned pain modulation.

**Table 4 jcm-11-01075-t004:** Association between CSI and symptom spread.

CSI	Median	Extramedian	*p*-Value
≥40	1	8	0.521
<40	4	17	

*p*-values reflect Fisher exact test. CSI: central sensitization inventory.

**Table 5 jcm-11-01075-t005:** Correlations between CSI and emotional wellbeing.

	CTS	
	Pearson Correlation Coefficient	Unadj *p*-Value
CSI with BECK	0.426	0.019 *
CSI with STAI	0.026	0.893
CSI with TSK	0.109	0.566

* reflects *p*-value that remains significant after Benjamini–Hochberg correction.

## Data Availability

The data presented in this study are available on request from the corresponding author.

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
