# Peer review of "Signs Indicative of Central Sensitization Are Present but Not Associated with the Central Sensitization Inventory in Patients with Focal Nerve Injury"

_jcm, 2022, doi:10.3390/jcm11041075_

Round 1

Reviewer 1 Report

Authors present a study on 30 patients with carpal tunnel syndrome (CTS) with indication for surgery (on a surgical waiting list), which was compared with 30 age-gender matched controls in matters of changes to central pain processing  established through psychophysical sensory testing
(bilateral pressure pain thresholds (PPT), conditioned pain modulation, temporal summation), pain distribution on body charts, pain severity and function questionnaires, psychological questionnaires and the central sensitization inventory (CSI). Authors report that  patients with
CTS have lower PPTs over the carpal tunnel bilaterally, reduced conditioned pain modulation efficacy, no
no differences in temporal summation and no association of CSI with
psychophysical measures or pain distributions indicative of altered central pain processing, but with a a correlation of the CSI with the Beck Depression Inventory.

Main limitation of the study is the low number of patients, however since there is a control group, some conclusions can be drawn. All patients were on a surgery waiting list - so I would change the title and the intention of the study accordingly - we do not know if the patients with moderate CTS, which is on conservative treatment, has the same parameters compared to healthy controls. Please clarify if the patients were on the neurosurgical, surgical, orthopaedic or plastic surgery - surgery waiting list.

Authors state that there is a spanish version of CSI.  I suggest to expand the introduction and discussion on CSI and provide more information on how the test is being performed. I suggest to include following studies for Discussion:

  1. Mikkonen J, Luomajoki H, Airaksinen O, Neblett R, Selander T, Leinonen V. Cross-cultural adaptation and validation of the Finnish version of the central sensitization inventory and its relationship with dizziness and postural control. BMC Neurol. 2021;21(1):141. Published 2021 Mar 31. doi:10.1186/s12883-021-02151-6
  2. Mayer TG, Neblett R, Cohen H, Howard KJ, Choi YH, Williams MJ, Perez Y, Gatchel RJ. The development and psychometric validation of the central sensitization inventory. Pain Pract. 2012 Apr;12(4):276-85. doi: 10.1111/j.1533-2500.2011.00493.x. Epub 2011 Sep 27. PMID: 21951710; PMCID: PMC3248986.
  3. Bennett EE, Walsh KM, Thompson NR, Krishnaney AA. Central Sensitization Inventory as a Predictor of Worse Quality of Life Measures and Increased Length of Stay Following Spinal Fusion. World Neurosurg. 2017 Aug;104:594-600. doi: 10.1016/j.wneu.2017.04.166. Epub 2017 May 4. PMID: 28479522. - please include and comment on this study, as it seems to be the only study which involves neurosurgical patients  
  4. Plazier M, Ost J, Stassijns G, De Ridder D, Vanneste S. Pain characteristics in fibromyalgia: understanding the multiple dimensions of pain. Clin Rheumatol. 2015 Apr;34(4):775-83. doi: 10.1007/s10067-014-2736-6. Epub 2014 Jul 22. PMID: 25048743.

Author Response

Authors present a study on 30 patients with carpal tunnel syndrome (CTS) with indication for surgery (on a surgical waiting list), which was compared with 30 age-gender matched controls in matters of changes to central pain processing  established through psychophysical sensory testing (bilateral pressure pain thresholds (PPT), conditioned pain modulation, temporal summation), pain distribution on body charts, pain severity and function questionnaires, psychological questionnaires and the central sensitization inventory (CSI). Authors report that  patients with CTS have lower PPTs over the carpal tunnel bilaterally, reduced conditioned pain modulation efficacy, no differences in temporal summation and no association of CSI with psychophysical measures or pain distributions indicative of altered central pain processing, but with a a correlation of the CSI with the Beck Depression Inventory.

Main limitation of the study is the low number of patients, however since there is a control group, some conclusions can be drawn. All patients were on a surgery waiting list - so I would change the title and the intention of the study accordingly - we do not know if the patients with moderate CTS, which is on conservative treatment, has the same parameters compared to healthy controls. Please clarify if the patients were on the neurosurgical, surgical, orthopaedic or plastic surgery - surgery waiting list.

We would like to thank the reviewer for his comments. We have further clarified the type of waiting list in material and methods. We have also considered changing the title; however, it is already rather long. We have therefore decided to specify the patient population further in the abstract.

The following changes were made to the Abstract: “Methods: Thirty healthy volunteers were matched with 30 people with unilateral CTS from the orthopaedic waitlist"

Methods section: "2.1. Participants: Thirty patients with unilateral CTS were recruited from Hand and Elbow surgery units from 12 Octubre Hospital and Severo Ochoa Hospital, both located in Madrid. All patients were on the  orthopaedic surgery waitlist"

Authors state that there is a Spanish version of CSI.  I suggest to expand the introduction and discussion on CSI and provide more information on how the test is being performed. I suggest to include following studies for Discussion:

We have further expanded on the CSI in the methods section and also incorporated most references suggested by the reviewer.

Methods: “The Central Sensitization Inventory (CSI) is a tool originally designed to identify patients with ‘central sensitivity syndrome’ (14).  It includes a wide range of 25 questions covering pain and stiffness, daily function, psychological factors (e.g., anxiety, depression), fatigue and memory.”

  1. Mikkonen J, Luomajoki H, Airaksinen O, Neblett R, Selander T, Leinonen V. Cross-cultural adaptation and validation of the Finnish version of the central sensitization inventory and its relationship with dizziness and postural control. BMC Neurol. 2021;21(1):141. Published 2021 Mar 31. doi:10.1186/s12883-021-02151-6

We have followed the reviewer’s advice and cited this article in the introduction as well as in the discussion.

  1. Mayer TG, Neblett R, Cohen H, Howard KJ, Choi YH, Williams MJ, Perez Y, Gatchel RJ. The development and psychometric validation of the central sensitization inventory. Pain Pract. 2012 Apr;12(4):276-85. doi: 10.1111/j.1533-2500.2011.00493.x. Epub 2011 Sep 27. PMID: 21951710; PMCID: PMC3248986.

We have added this citation in the introduction to explain the origin and development of the CSI questionnaire.

  1. Bennett EE, Walsh KM, Thompson NR, Krishnaney AA. Central Sensitization Inventory as a Predictor of Worse Quality of Life Measures and Increased Length of Stay Following Spinal Fusion. World Neurosurg. 2017 Aug;104:594-600. doi: 10.1016/j.wneu.2017.04.166. Epub 2017 May 4. PMID: 28479522. - please include and comment on this study, as it seems to be the only study which involves neurosurgical patients  

We have included this reference in the introduction to contextualise the use of the CSI questionnaire.

  1. Plazier M, Ost J, Stassijns G, De Ridder D, Vanneste S. Pain characteristics in fibromyalgia: understanding the multiple dimensions of pain. Clin Rheumatol. 2015 Apr;34(4):775-83. doi: 10.1007/s10067-014-2736-6. Epub 2014 Jul 22. PMID: 25048743.

We have decided not to include this last suggestion as this article focuses on fibromyalgia which is not the main topic of our article.  

Reviewer 2 Report

The manuscript “Signs Indicative of Central Sensitization Are Present but not Associated with the Central Sensitization Inventory in Patients with Focal Nerve Injury” by Luis Matesanz-García, Ferran Cuenca-Martínez, Ana Isabel Simón, David Cecilia, Carlos Goicoechea- García, Josué Fernández-Carnero and Annina B. Schmid” aimed to identify alterations in central pain mechanisms in patients with CTS using psychophysical sensory testing and pain mapping and to investigate whether the CSI is associated with psychophysical parameters and pain distributions indicative of central pain mechanisms.

Below are my comments and remarks regarding the article:

1. Lack of photos of sample measurements
2. Was an electrophysiological examination of the healthy side performed?
3. Cervical spine and upper limb diagnosis what does it mean? - whether the cervical spine or neuropathy was examined by electrophysiological examination or whether it was excluded only on the basis of the lack of diagnosis.
4. Reference not compliant with mdpi requirements

Author Response

The manuscript “Signs Indicative of Central Sensitization Are Present but not Associated with the Central Sensitization Inventory in Patients with Focal Nerve Injury” by Luis Matesanz-García, Ferran Cuenca-Martínez, Ana Isabel Simón, David Cecilia, Carlos Goicoechea- García, Josué Fernández-Carnero and Annina B. Schmid” aimed to identify alterations in central pain mechanisms in patients with CTS using psychophysical sensory testing and pain mapping and to investigate whether the CSI is associated with psychophysical parameters and pain distributions indicative of central pain mechanisms.

  1. 1. Lack of photos of sample measurements

Response: We would like to thank Referee 2 for reviewing our manuscript. Following your suggestion, we have added photos of the three psychophysical measurements as a supplementary Figure. As these are well established measures, we do not think this Figure needs to be in the main text. However, we are open to include the figure in the main text if the Editors wish so.  

  1. Was an electrophysiological examination of the healthy side performed?

Response: As per routine practice in participating hospitals, electrodiagnostic testing was only performed on the symptomatic (affected) side. Your comment is however important, as sometimes there can be a subclinical CTS even on a pain free side. We have further clarified in the methods that electrodiagnostic testing was only performed on the symptomatic side. We have also added the possibility of subclinical CTS in the limitations section.

Methods: "had positive Tinel’s and Phalen’s sign and had electrodiagnostic confirmation of moderate to severe CTS on the affected side according to the American Association of Neuromuscular and Electrodiagnostic Medicine (25)."

Discussion: " As per routine practice in participating hospitals, the electrodiagnostic test was only performed on the affected side.  Subclinical cases of CTS on the contralateral side may therefore have been missed (62).”

  1. Cervical spine and upper limb diagnosis what does it mean? - whether the cervical spine or neuropathy was examined by electrophysiological examination or whether it was excluded only on the basis of the lack of diagnosis.

Response: The electrodiagnostic testing included standard surface recordings for the median, ulnar and radial nerve. Needle EMG was performed if there were indications for nerve root involvement. We have clarified this in the methods section:  

“Patients were excluded if the electrodiagnostic testing identified sensory and/or motor deficits of the radial and/or ulnar nerve, if any indication for nerve root involvement was present (e.g., needle EMG) or”

  1. Reference not compliant with mdpi requirements

Response:  We apologise for this omission, which we have now corrected.

Round 2

Reviewer 1 Report

The authors have sufficiently answered to remarks of the reviewer.